# A network-based analysis detects cocaine-induced changes in social interactions in *Drosophila melanogaster*

**Milan Petrović**[1,2]*, **Ana Meštrović**[1,2], **Rozi Andretić Waldowski**[3], **Ana Filošević Vujnović**[3]

**1** Department of Informatics, University of Rijeka, Rijeka, Croatia, **2** Center for Artificial Intelligence and Cybersecurity, University of Rijeka, Rijeka, Croatia, **3** Department of Biotechnology, Laboratory for behavioral genetics, University of Rijeka, Rijeka, Croatia

* milan.petrovic@inf.uniri.hr

## Abstract

Addiction is a multifactorial biological and behavioral disorder that is studied using animal models, based on simple behavioral responses in isolated individuals. A couple of decades ago it was shown that *Drosophila melanogaster* can serve as a model organism for behaviors related to alcohol, nicotine and cocaine (COC) addiction. Scoring of COC-induced behaviors in a large group of flies has been technologically challenging, so we have applied a local, middle and global level of network-based analyses to study social interaction networks (SINs) among a group of 30 untreated males compared to those that have been orally administered with 0.50 mg/mL of COC for 24 hours. In this study, we have confirmed the previously described increase in locomotion upon COC feeding. We have isolated new network-based measures associated with COC, and influenced by group on the individual behavior. COC fed flies showed a longer duration of interactions on the local level, and formed larger, more densely populated and compact, communities at the middle level. Untreated flies have a higher number of interactions with other flies in a group at the local level, and at the middle level, these interactions led to the formation of separated communities. Although the network density at the global level is higher in COC fed flies, at the middle level the modularity is higher in untreated flies. One COC specific behavior that we have isolated was an increase in the proportion of individuals that do not interact with the rest of the group, considered as the individual difference in COC induced behavior and/or consequence of group influence on individual behavior. Our approach can be expanded on different classes of drugs with the same acute response as COC to determine drug specific network-based measures and could serve as a tool to determinate genetic and environmental factors that influence both drug addiction and social interaction.

## Introduction

Drug addiction is a complex neurological disorder that is influenced by interactions between brain circuits, genetics, the environment, and individual social experiences [1]. Addictive

**Data Availability Statement:** The data underlying the results presented in the study are available

from https://github.com/milanXpetrovic/my_module.

**Funding:** This work was supported by the Croatian Science Fondation grant IP-2018-01-2794 to RAW and University of Rijeka project uniri-mladi-intpo-20-38 to AFV. The funders had no role in study design, data collection and analysis, decision to publish, or preparation of the manuscript.

**Competing interests:** The authors have declared that no competing interests exist.

drugs such as cocaine, methamphetamine, nicotine and ethanol, are a chemically heterogeneous group with very distinct molecular targets [2]. Common to all drugs of abuse is an increase in the monoaminergic concentration, primary dopamine (DA) in the mammalian ventral tegmental area (VTA), frontal cortex and nucleus accumbens [3]. Drugs enter the brain and bind to initial protein targets: dopamine transporter on the outer membrane of the neuron, and the vesicular monoamine transporter inside the neuron. These events perturb synaptic transmission, which in turn causes acute behavioral effects of the drug measured as increase in locomotion. Acute effects of the drug cannot explain addiction by themselves, since addiction is the result of functional and morphological changes in the brain that develop as a consequence of the physiological and behavioral adaptation to the drug [4].

The combination of the distinct overlapping behavioral changes, which may have distinct polygenic etiologies influenced by genes pleiotropy, makes it difficult to assignee, separate or detect the roles of environmental cofactors in determining addiction phenotypes. Thus, study of simple to more complex behaviors induced by different drugs is possible using animal models [4]. Many of the discrete behavioral aspects of addiction have direct relevance to the development of addiction in humans. Under laboratory conditions, drug induced change in the behavior is studied on individual animals by quantifying discrete aspects of addiction behavior such as sensitivity to acute dose in drug naïve animals, oral preferential consumption, tolerance, conditional place preference, relapse and consumption despite negative consequences [5]. The results are presented as an average of isolated behavioral scores. This approach masks individual differences in order to perform more objective and reproducible quantification of drug induced phenotypes, and new lines of evidence now associate individual social experience [6] and influence of the group with the simple behaviors induced by drugs or predisposition to develop addiction [7, 8]. It is important to note that, the higher the complexity of the behavior which is induced and measured, the greater the relevance of the model for human study.

Genetic homology with humans, and translational potential for biomedical research have promoted *Drosophila melanogaster* as a model organism for a range of human diseases, including addiction [9]. *Drosophila*, as with vertebrate models, can be used for behaviors induced by non-voluntary [10, 11] and voluntary [12, 13] psychostimulant administration. Like humans, flies become hyperactive and disinhibited upon exposure to low doses of psychostimulant, uncoordinated at moderate doses, and sedated at high doses. Each of *Drosophila*'s well-established genetic and behavioral approaches give different information about the addiction phenotype of interest, and can be used in genetic screens or neuronal citrus analysis [14], but without dissection of social experience or influence of the group on the individual behavioral response. Analyzing individuals within a group can provide information on behavioral responses of all individuals under different conditions, but without providing information about group structure and possible influence of the group on individual behavior.

Social interaction network analysis, based on interactions between individuals in a group, has been used successfully to gain insights into the formation and function of group structure [15, 16]. This type of analysis, based on the graph theory, can be used to investigate transmission processes in a group, which is a basis for complex phenomena such as social grooming, decision-making, and understanding of hierarchy. Effectiveness of task allocation and information flow in social insects was previously determined using SINs [17]. The authors of those studies argue that network-based analysis is an efficient method for analysis of complex collective behaviors. Although *Drosophila* does not satisfy the strict definition of a social insect (it does not live in colonies and does not show division of labor) it does show distinct group dynamic that has become recognized as another behavioral phenotype [18–20]. Individual behaviors and social network analysis highlight different aspects of social interaction, and are

complementary for understanding group behavior since even simple pairwise interactions can generate complex group dynamics.

*Drosophila* proves to be a useful model organism for mechanistically exploring complex behaviors, but only a small number of studies have examined social group interaction in flies. Video recording and tracking of freely moving flies in the group has led to the identification of social interaction criteria and indicated the importance of chemo-sensory cues in regulating distance and interaction between flies [21]. There have been a number of studies focused on identifying neurogenetics of group behavior [22–25], environmental influence on modulation of group behavior [26–29] and evolution and inheritance of social behavior in *Drosophila* [30, 31]. These studies showed that social behavior is another complex behavioral phenotype that needs to be further characterized in the context of genetic and environmental manipulations, especially with regard to addiction.

The most commonly used program for the automated transformation of video data into numerical values for social interaction analysis is Ctrax, which enables tracking up to 50 flies [32, 33]. Although often used, Ctrax can lead to errors when tracking individuals, due to the loss of tracking or swapping of intensities, so new software has been developed that eliminates these problems [34]. With manual or automatic annotation, it is possible to track individuals within a group and to identify specific types of behavior [35]. We used Flytracker [35] for tracking the flies in the first step of our experiment. In the next step we used a library called NetworkX, written in the Python programming language, and implement all data processing and network-based analysis. Thus we were able to automatically extract and analyze SINs data from 10 minute videos of 30 individuals [36]. For network visualisations we used Gephi, an open source software [37]. We describe a novel network-based methodology for the analysis of social behavior in adults *D. melanogaster*. Within the methodology we propose a set of network-based measures that quantify social interactions of *D. melanogaster* represented via Social Interaction Networks (SINs). This methodology aims to analyze the behavior induced by psychostimulants by analyzing SINs structures on the local, middle and global levels. It enables identifying structural properties and differences in *D. melanogaster* populations. Proposed approach provides insight into parameters and characteristics of group behavior in presence and absence of psychostimulants that could be of interest for further bioinformatics research studies and analyses.

## Materials and methods

### Fly strain and breeding

We used *wild type* (*wt*) flies of *CantonS* background raised on cornmeal food (S1 Protocols). The flies were bred in an incubator at 25˚C and 70% humidity with a 12h light/12h dark cycle (S1 Protocols). One day before the video recording, we collected two groups of 30 adult male flies, 3–5 days old using a microscope and $CO_2$ anesthesia (S1 Protocols). One population was transferred to regular molasses food (S1 Protocols) without (CTRL) or with 0.50 mg/mL cocaine-hydrochloride (COC, Sigma Aldrich, HPLC grade) (S1 Protocols). Each group in the population had 30 individuals, and in total 9 CTRL and 11 COC groups were tested. Flies were then placed in the incubator for 24 hours under the same conditions as for breeding until recording (Fig 1A).

### Arena and experiment setup

Video recording of approximately 30 adult flies in each group was performed in an open field circular arena developed in collaboration with the University of Rijeka Faculty of Engineering, Department of Mechanical Engineering and ID products development. The arena was 3

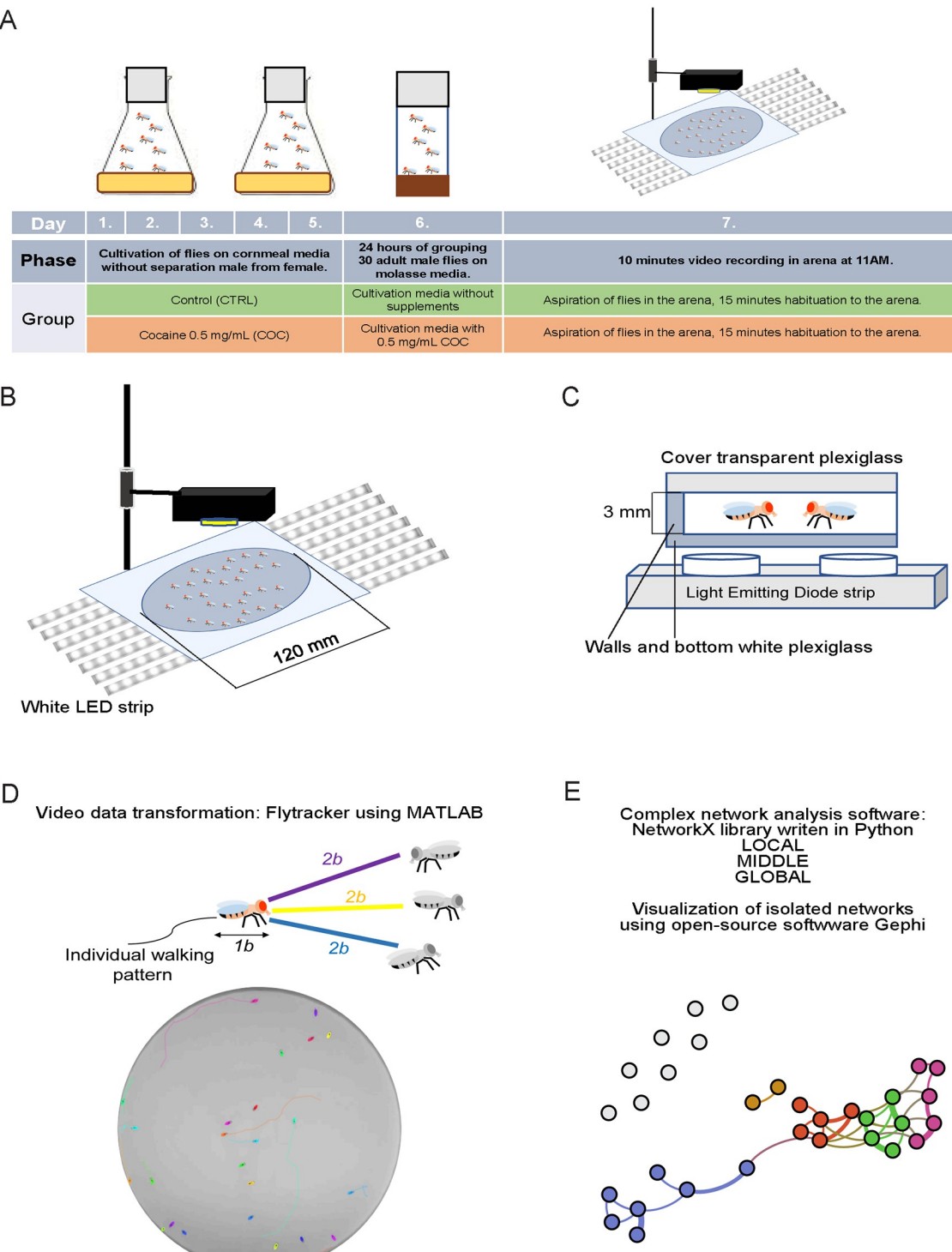

**Fig 1. Experimental setup. A)** Timeline of the experiment. **B)** Graphical representation of the video recording setup. 30 flies were recorded for 10 minutes in a 120 mm diameter circular arena with a bottom made of white Plexiglas illuminated with a white LED strip. **C)** Cross section of the arena. The outer cover of the arena is made of transparent Plexiglas placed at a height of 3 mm, to allow movement of flies in two dimensions only. **D)** Distance interaction threshold and possible types of interactions between two flies extracted from the video recording shown in the circle. The size of one fly is considered as one body length (1b) which is approximately 2 mm. An interaction threshold of two body lengths (2b) or 4 mm was used where one fly (the interactor fly shown in color) approaches another fly (the interacted fly shown in gray) in various orientations. Possible interactions are: head-to-head

(violet arrow), head-to-tail (yellow arrow) or head-to-body (blue arrow). Data were extracted using Flytracker, which is written in MATLAB. Measured interactions are not oriented, but there is an additional threshold of the length of the interaction of a minimum of 0.5 seconds. **E)** Data extracted using Flytracker was used to analyze and visualize social interactions between flies at a local, middle, and global level. A package, written in the Python programming language, was used to analyze network data. For visualization and editing, we used the open-source software Gephi. All data and the Python package is publicly available at https://github.com/milanXpetrovic/my_module.

millimeters high in order to restrict movement to two dimensions, and 120 millimeters in diameter to allow movement of the flies without crowding [26] (Fig 1B and 1C). The bottom was white translucent Plexiglas back-lit with a white LED strip, and the top was a transparent Plexiglas cover with a height of 2 mm (Fig 1B). To avoid the influence of the time of the day on the behavior, video recording was performed each day at 11.00 AM [38]. For video recording, we used a Logitech C920HD Pro camera with 1920 x 1080 pixels video resolution and 24 frames per second. The flies were transferred from cultivation vials to the arena using an aspirator and left for 15 minutes to habituate [39]. To minimize the potential effect of fluctuation in temperature and humidity, recording lasted only 10 minutes and was performed on 9 CTRL and 11 COC groups. Between each experiment, the arena was washed and dried to eliminate the potential influence of olfactory cues [40].

## Tracking and annotation software

Fly tracking was done using the open source software Flytracker, developed using MATLAB. Each video frame was segmented to separate the flies from the black background. Each fly was detected as an area of approximately 20 pixels. Interactions between two flies were extracted without specifying the type of interactions, including differentiating between head-to-head from head-to-tail interaction (Fig 1D). Based on the data extracted using Flytracker, we constructed a set of SINs to analyze and visualize social interactions between flies (Fig 1E).

## Transformation of video data into social interaction networks (SIN)

Social interaction networks are represented as weighted graphs $G = (V, E)$, a pair of two data sets. The first, $V$, consists of nodes (vertices) that represent files, and the second, $E$, consists of links (edges), with associated weights, which quantify interactions between flies. We introduced two types of weight factors: the number of interactions and the total duration of these interactions: **(i) number (count) of interactions between two flies (nodes)** and **(ii) total duration of all interactions between two flies (nodes)**. More precisely, all interactions between two files are represented using only one link, however, information about the number of different interactions and the total duration of the interactions is captured as the weight of this link. In this way, the weights of links in the network are determined for each fly in the group, as the number of times that fly interacted with other flies in the group during the 10 minutes of the video (number of interactions), and as the total time that these two flies spent in all their interactions (duration of interactions).

Throughout the data analysis, each node in the network is assigned a unique identity for each fly from the group of 30 flies in the CTRL and COC population, and it remains unchanged throughout the entire video. Visual representation of the network is constructed with each fly as a node and the link representing the interaction between two flies (nodes), defined as when two flies remain within two body lengths (4mm) of each other for a duration of longer than 0.5 seconds. The criteria of distance and time threshold for social interaction were taken from a previously published related study [21].

We constructed 11 networks for COC groups of flies and 9 networks for CTRL groups of flies. We performed network analyses and comparisons of the COC and CTRL networks at the global, local, and middle levels. A package written in the Python programming language, named NetworkX, was used to analyze network data [41]. For network visualization and editing we, used an open-source software Gephi [37]. Statistical analysis was performed using independent-samples *t*-tests. These tests are Welch-corrected, since group sizes are different and with significance defined as $p < 0.05$.

## Characterization of SINs

We present SIN analysis through three categories: global, local, and middle network-based measures. In the following subsections, we provide definitions and explanations of the measures that we have identified as being important for the SIN characterization of CTRL and COC-fed populations.

**Global-level SINs measures.** Global network measures are of great importance for describing and characterizing different classes of networks [42, 43], and are described here.

**Average shortest path length** ($L$) represents the average number of stops needed to reach two distant flies (nodes) in the network. In the case of SINs, it measures the distance between flies in terms of the number of interactions. If $d_{ij}$ denotes the number of links lying on the shortest path between nodes $i$ and $j$, the average shortest path length is calculated as follows:

$$L = \sum_{i,j \in V} \frac{d_{ij}}{N(N-1)}. \tag{1}$$

**Network diameter** ($D$) is the longest of all the calculated shortest paths in a network:

$$D = \max(d_{ij}). \tag{2}$$

Based on distances between flies (nodes) in the network, the **global efficiency** of the network ($E_{glob}(G)$) is defined as a property related to the distance and connectedness of the flies (nodes) in the network. It provides the notion of how efficiently the information may flow through the network:

$$E_{glob}(G) = \frac{1}{N(N-1)} \sum_{i \neq j \in V} \frac{1}{d_{ij}}. \tag{3}$$

For all distance measures, if a network contains more than one component, the measure is calculated taking into account only the largest component. In our experiments we calculated properties of clustering. Based on the local clustering coefficient (explained in the next section) of a fly (node) it is possible to calculate an average clustering coefficient of a network as:

$$\langle c \rangle = \frac{1}{N} \sum_{i \in V} c_i. \tag{4}$$

**Network transitivity** ($T$) is the global measure of clustering, where possible triangles are identified by the number of triads (two interaction (links) with a shared fly (node)):

$$T = 3 \frac{\#triangles}{\#triads}. \tag{5}$$

**Network density** ($\rho$) is as the fraction of all links over the total possible number of interactions (links):

$$\rho = \frac{2K}{N(N-1)}. \tag{6}$$

**Degree heterogeneity** ($d_{het}$) represents the diversity in fly (node) degrees and the diversity in the structure of the network. It is calculated as a fraction of a standard deviation and average degree. This measure has a direct correspondence with the entropy of a complex network characterized by the standard Shannon's measure of information.

**Asortativity** ($r$) is a measure of a preference for attaching a fly (node) to other similar flies (nodes) The network shows assortative mixing by a degree if flies (nodes) tend to be connected to other flies (nodes) with a similar degree. This value is calculated as Pearson correlation:

$$r = \frac{\sum_{jk} jk(e_{jk} - q_j q_k)}{\sigma_q^2}, \tag{7}$$

where $e_{jk}$ is the joint probability distribution of the excessive degrees of the two flies (nodes) at either end of a randomly chosen interaction (link). Here $q_j q_k$ and $\sigma_q^2$ are the expected value or mean, and standard deviation, of the excess degree distribution.

**Local-level SINs measures.** Local level network-based measures are based on the number of fly (node) interactions (links), fly (node) position within the network, and the relationship with other flies (nodes). Some of these measures can be used for ranking flies in the network of interactions. Every local network measure can be applied for ranking flies by their central position in the network, and thus we often refer to these measures as centrality measures. The appropriate usage of centrality measures depends on understanding the domain and the type of link in the network [44].

In this study, we applied the following local level measures (the different node rankings are shown in Fig 2): degree centrality, weighted degree, closeness centrality, betweenness centrality, eigenvector centrality, information centrality, and local clustering coefficient.

**Degree centrality** ($dc_i$) of a fly (node) $i$ is the measure that takes into account total number of interaction (links) incident with a fly (node) (Fig 2A). It is usually normalized by dividing it by the maximum possible degree $N - 1$:

$$dc_i = \frac{k_i}{N-1}. \tag{8}$$

In the context of SINs degree centrality of a fly may be described as the number of interactions of this fly with other flies in the network [45].

In weighted networks a weighted degree $s_i$ is refereed to as **node strength**. Strength for a fly (node) $i$ is defined as the sum of all weights attached to links belonging to this node:

$$s_i = \sum_{j \in \Pi(i)} w_{ij}, \tag{9}$$

where $\Pi(i)$ denotes set of neighbouring nodes of a node $i$.

In this study, we analyze two different types of strengths based on two different types of weights described in the previous section: **node strength based on the number of fly interactions** and **node strength based on the duration of fly interactions**.

The **closeness centrality** ($cc_i$) of a fly (node) reflects how close a fly (node) is to all other flies (nodes) in the network (Fig 2B), and is calculated as the average of the shortest path length from the fly (node) to each other fly (node) in the network. The shortest path between two flies

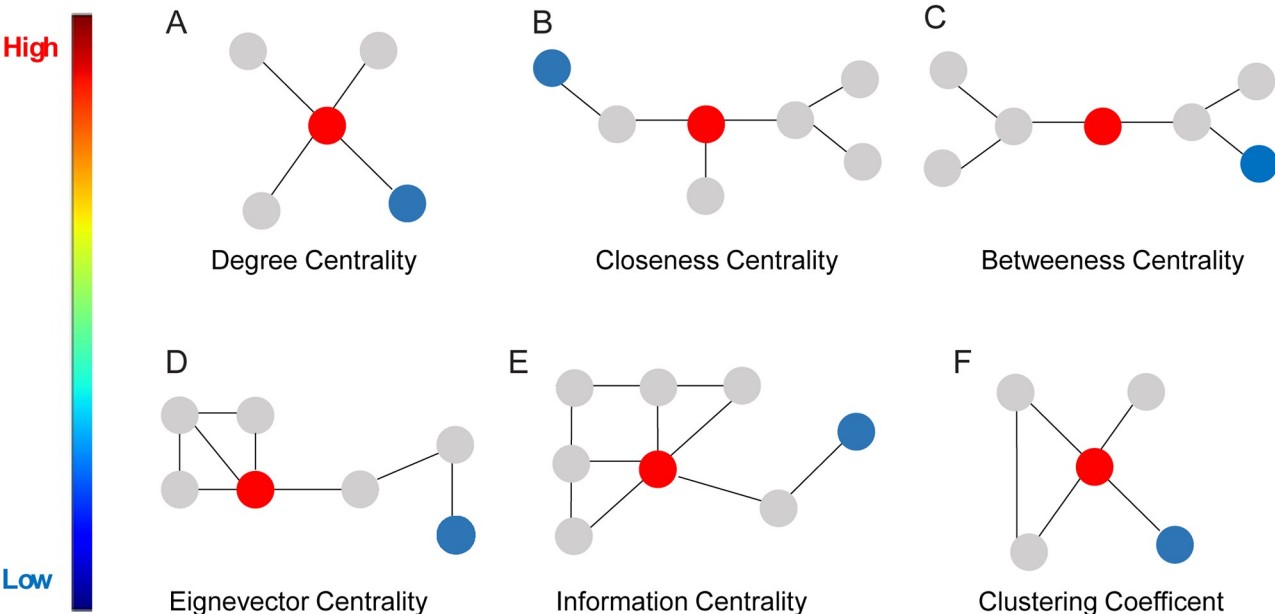

**Fig 2. Local SINs measures.** Visual representation of differences between local network-based measures in terms of the node centrality. Red colored nodes are nodes with the highest value of the centrality measure, while blue colored are nodes with lower values of the centrality measure. This property varies and depends on the chosen centrality network measure.**A)** Degree centrality or number of individual interactions (links) with other flies (nodes) in the network. **B)** Closeness centrality measure defines how close a fly (node) is to all other flies (nodes) in the network. **C)** Betweenness centrality detects the influence of a fly (node) on the flow of information, depicted as a bridge from one part of a network to another. **D)** Eigenvector centrality measures the influence a fly (node) in a network based to its neighbours. **E)** Information centrality or average information of all paths originating from a fly (node). **F)** Clustering coefficient defines how well are neighbouring flies interconnected.

is the least number of interactions needed to connect these two flies. In the context of SINs, a fly with a higher value of closeness centrality has close interactions with all other flies in the network. For closeness centrality let $d_{ij}$ be the shortest path between flies (nodes) $i$ and $j$. The normalised closeness centrality of a fly (node) $i$ is given by:

$$cc_i = \frac{N-1}{\sum_{i \neq j} d_{ij}}. \tag{10}$$

**Betweenness centrality** ($bc_i$) quantifies the number of times a fly (node) acts as a bridge along the shortest path between two other flies (nodes) (Fig 2C). A fly with a higher value of betweenness centrality has interactions with various different communities of flies in the network. Let $\sigma_{jk}$ be the number of shortest paths from fly (node) $j$ to fly (node) $k$ and let $\sigma_{jk}(i)$ be the number of those paths that pass through the fly (node) $i$. The normalized betweenness centrality of a fly (node) $i$ is given by:

$$bc_i = \frac{\sum_{i \neq j \neq k} \frac{\sigma_{jk}(i)}{\sigma_{jk}}}{(N-1)(N-2)}. \tag{11}$$

**Eigenvector centrality** is a measure that takes into account the centrality of adjacent flies (nodes) (Fig 2D). Relative scores are assigned to all flies in the network based on the concept that connections to high-scoring flies (nodes) contribute more to the score than equal connections to low-scoring flies. A high eigenvector score means that a fly has interactions with many flies who themselves have high scores. For the fly (node) $i$ and constant λ centrality $ce_i$ of fly

(node) *i* is defined as:

$$ec_i = \frac{1}{\lambda} \sum_{j \in \Pi(i)} ec_j. \tag{12}$$

**Information centrality** ($ic_i$) of a fly (node) is calculated as an average of the information of all paths originating from that fly (node) (Fig 2E). This measure is also known as current-flow closeness centrality. The measure is similar to closeness centrality, however it it takes into account all the paths, not only the shortest paths. The information centrality for a fly (node) *i* is given by:

$$ic_i = \frac{N}{\sum_{j \in N} \frac{1}{I_{ij}}}, \tag{13}$$

where $I_{ij}$ is the centrality of a path from fly (node) *i* to *j*.

In the context of SINs, high scores in information centrality of a fly implies that it has more interaction to other files that have large number of interactions.

**Local clustering coefficient** ($C_i$) of a fly (node) measures how well are neighbors interconnected and quantifies if they are becoming a clique i.e. a subgraph with flies (nodes) all connected with each other (Fig 2F). The local clustering coefficient is calculated as the proportion of interactions (links) between the flies (nodes) within its neighborhood divided by the number of interactions (links) that could possibly exist between them. Real-world networks (and in particular social networks) have on average higher clustering coefficient than random networks (when comparing networks of the same size). The clustering coefficient of a fly (node) *i* is defined as:

$$C_i = \frac{e_{ij}}{k_i(k_i - 1)}, \tag{14}$$

where $e_{ij}$ represents the number of pairs of neighbours of a fly (node) *i* that are connected.

**Middle-level SINs measures.**   At the middle level of network analysis, focus is on connections between flies (nodes) of a smaller groups, usually named sub-graphs or communities. A number of flies (nodes) in a sub-graph is smaller than the total number of flies in the network, and size is often predefined by certain rules or algorithms.

Communities are groups of densely interconnected flies (nodes) within a network. Flies (nodes) in a community have a greater amount of connections amongst each other than with other flies (nodes) in the network. Several algorithms are used for community detection such as hierarchical clustering, GirvanNewman's algorithm, minimum-cut method and others. One of the most efficient is the Louvain algorithm [46]. Louvain algorithm is based on a greedy optimization method that optimizes the modularity of a network's partitions.

**Modularity** (*Q*) depicts the quality of the partitioning into communities. The value of the modularity is in the range [−0.5, 1]. Networks with higher values have dense connections between the flies (nodes) within community, but sparse connections between flies (nodes) in different communities. Let $e_{ij}$ be the fraction of interactions (links) in the network that connect flies (nodes) in group *i* to those in group *j* and let $a_i = \Sigma_j e_{ij}$. Then the modularity can be calculated using the following equation:

$$Q = \sum_{i=1}^{N} (e_{ii} - a_i^2). \tag{15}$$

Social networks are usually formed in the way that there is one large connected component called largest connected component and certain number of smaller separate components. We have analyzed the number of connected components $N_C$ within the community because it directly affects the number of communities.

## Results

### Validation of the network-based methodology using global network-based measures

Comparison of the network-based global level measures of cocaine-fed (COC) and control (CTRL) populations enables validation of the proposed network-based methodology. Previous studies in flies showed a difference in locomotor activity induced by different COC concentrations compared to CTRL [10]. We confirmed that being fed with 0.50 mg/mL COC increases locomotion, but also affects global network-based measures (Table 1), with weighted attributes for several measures (S3 Table) including the statistical analysis (S6 Table). These results corroborate that there are also differences between SINs of COC and CTRL populations. COC fed flies have a higher average number of interactions between flies in the group (Table 1), but lower global measures of shorter path length and network diameter. These global network-based measures confirm the increasing network density in COC fed flies and higher values of the global efficiency measure compared to CTRL networks. Higher values of global efficiency are consequences of dense connections between flies fed with the COC. Average strength, based on the duration of interactions, is the most significantly affected measure between the two populations.

Using a heat map we were able to better visualize networks regarding arena shape and diameter, and determine the location of interactions. For each population, we used one COC and one CTRL group of flies (Fig 3). Both groups of flies tended to aggregate closer to the edges of the arena, a phenomenon previously described in *Drosophila* as centrophobism or thigmotaxis [47]. Although we did not perform additional statistical analysis, it appears that COC-fed flies show more interaction closer to the center of the arena, while CTRL flies tended to aggregate closer to the edges.

**Table 1. Global level network-based measures.** Table represent values calculated at the global level of network for control (CTRL) and cocaine (COC) populations. Usage of specific edge weight (count, duration) in the calculation of measure is indicated in brackets next to the name of the measure.

|  | mean CTRL | mean COC |
| --- | --- | --- |
| Number of nodes, $N$ | 31.11±0.65 | 28.00±0.82 |
| Number of links, $K$ | 55.44±9.10 | 59.91±8.17 |
| Average degree, $<k>$ | 3.57±0.59 | 4.24±0.54 |
| Average strength (count), $<s>$ | 4.94±0.85 | 7.08±1.09 |
| Average strength (duration), $<s>$ | 7.78±1.13 | 21.93±4.50 |
| Average number of links, $<l>$ | 1.79±0.30 | 2.12±0.27 |
| Network density, $\rho$ | 0.12±0.02 | 0.16±0.02 |
| Avg. shortest path length, $L$ | 2.58±0.13 | 2.16±0.05 |
| Diameter, $D$ | 5.78±0.36 | 4.55±0.23 |
| Global efficiency, $E_{glob}$ | 0.33±0.05 | 0.39±0.04 |
| Transitivity, $T$ | 0.24±0.04 | 0.31±0.02 |
| Heterogeneity, $d_{het}$ | 0.80±0.07 | 0.80±0.09 |
| Assortativity, $r$ | −0.10±0.06 | -0.12±0.04 |

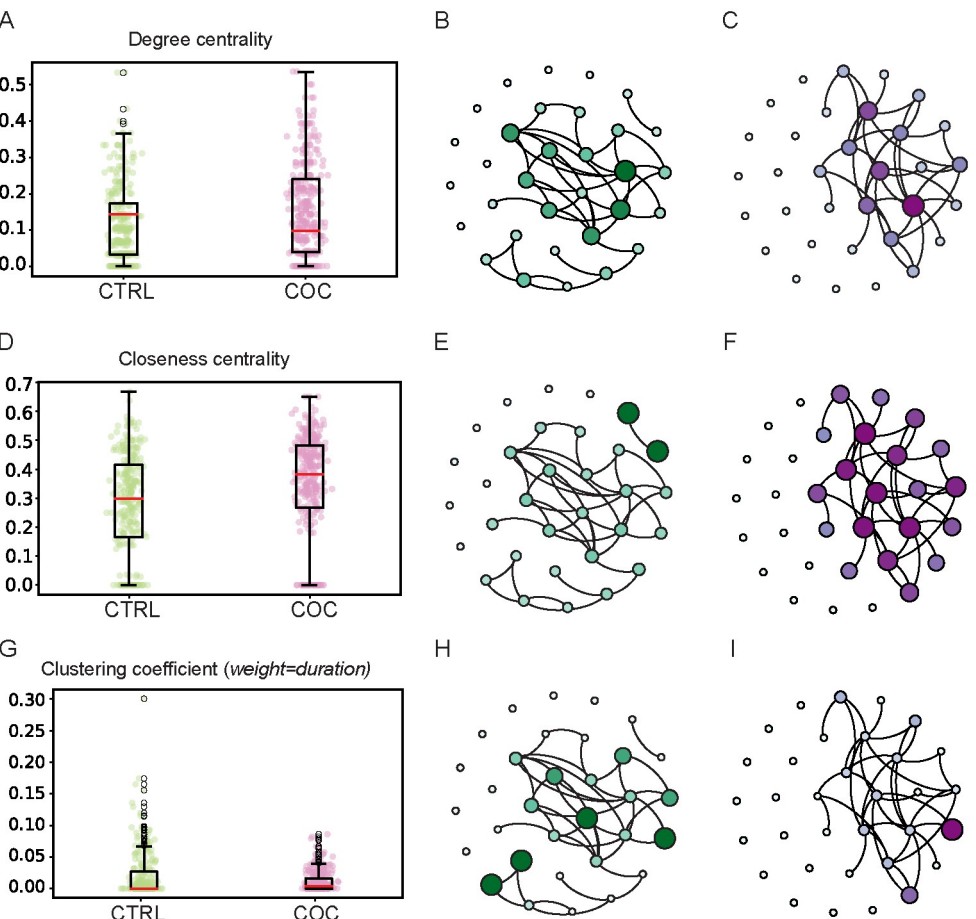

**Fig 3. COC and CTRL heat maps of location of interactions.** Visualization shows a graphical representation of the retention duration at a particular location in the arena during experiment. The visualization was made for populations in total where **A)** shows COC population and **B)** CTRL population.

## Difference in the spacing and interactions among flies between COC and CTRL networks

To describe the behavior of individual flies within a group, we measured SINs at the local level and found significant differences between CTRL and COC fed populations in terms of degree centrality, closeness centrality and clustering coefficient, information centrality, and strength distribution (Fig 2) (p-values in S6 Table). Degree centrality, closeness centrality, and clustering coefficient are depicted on box plot diagrams as a median of nine CTRL and eleven COC groups (Fig 4A, 4D and 4G). To graphically illustrate the differences in the SIN structure we selected representative CTRL and COC groups for: degree centrality (Fig 4B and 4C), closeness centrality (Fig 4E and 4F) and clustering coefficient (Fig 4H and 4I).

There is a higher average degree of centrality in the CTRL groups relative to COC-fed fly groups (Fig 4A, S1 Fig) indicating that more flies are interacting (degrees) with other flies (nodes). Thus untreated flies interact with other flies in the group, while COC feeding leads to more isolation (Fig 4B and 4C). The untreated CTRL flies form subgroups or communities that communicate with each other, while COC fed flies interact in a large group. The number of popular flies, those with a large number of interactions with others, is similar in CTRL and

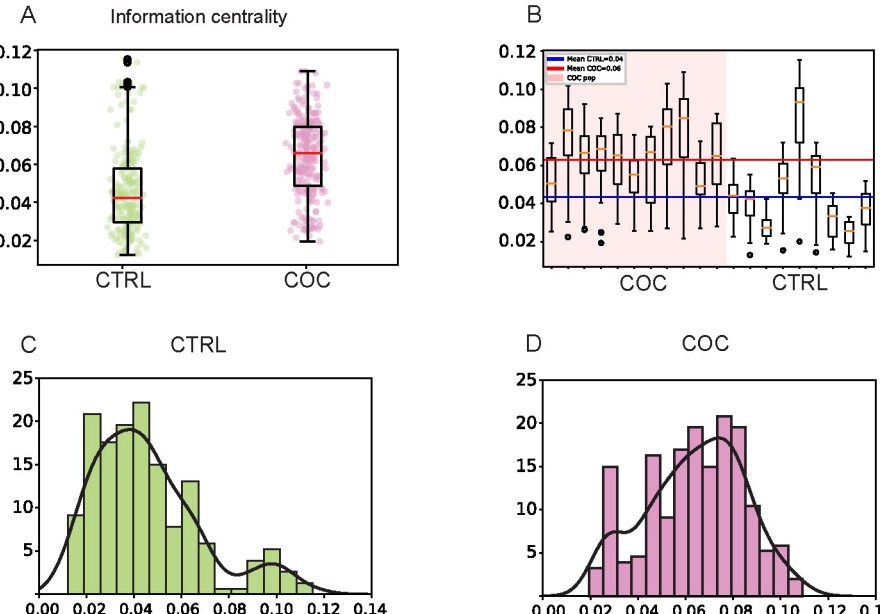

**Fig 4. Differences between CTRL and COC SINs in degree centrality, closeness centrality, and clustering coefficient. A), D), G)** Box plot of median values for nine CTRL SINs (n = 270 flies) grouped on the regular food and eleven COC SINs (n = 330) grouped and orally administrated to 0.50 mg/mL of cocaine for 24 hours before tracking in degree centrality A), closeness centrality D) and clustering coefficient G). Data are extracted from 10 minute videos using FlyTracker and analyzed using NetworkX. Statistical analysis was performed using independent-samples *t*-tests with Welch-correction since group sizes are different. *p*- values less than 0.05 are taken as significant and presented in S1 Table. **B), E), H)** Graphical illustration of local-level SINs from the isolated CTRL group using an open-source software Gephi for visualization and editing of the visual appearance of a given network. Nodes are represented with size and color proportional to the values of their centrality measures. Nodes with higher values of centrality measure degree centrality B), closeness centrality E) and clustering coefficient H) are shown as bigger and darker. **C), F), I)** Graphical illustration of local-level SINs from the isolated COC group using an open-source software Gephi for visualization and editing of the visual appearance of a given network. Nodes are depicted with the size and color proportional to the values of their centrality measures. Nodes with higher values of centrality measure degree centrality C), closeness centrality F) and clustering coefficient I) are shown as bigger and darker.

COC populations as indicated by the size of the circles and the intensity of coloration (Fig 4B and 4C).

In the case of measures based on weights, we test different possibilities of weights: number of interactions (weight = count), and duration of interactions (weight = duration).

Clustering coefficient values support the degree centrality medians observed since it shows that neighboring flies (nodes) are connected and if they are becoming a clique. CTRL population have higher clustering coefficients (Fig 4G and 4H) and they form subgroups with popular flies (nodes) (Fig 4H). These observations are generated using the weight for the duration and there were no statistically significant differences with weight set as count (Fig 2, Suppl. Figs S10 and S11).

Average closeness centrality, which describes how close a fly (node) is to other flies (nodes) in the network, is significantly higher for SINs in COC-fed groups (Fig 4D, S3 Fig) than in CTRL groups, and this is evident in the graphical representation as a close spacing of the flies (nodes) within a single group (Fig 4F). As a consequence, there are more isolated individuals without connections with other flies. In contrast, untreated flies have more flies (nodes) with higher distances to others in the network, but also fewer isolated individuals (Fig 4E).

Another measure that was significantly different between our groups was information centrality (Fig 5), which represents the average information of all paths originating from a given

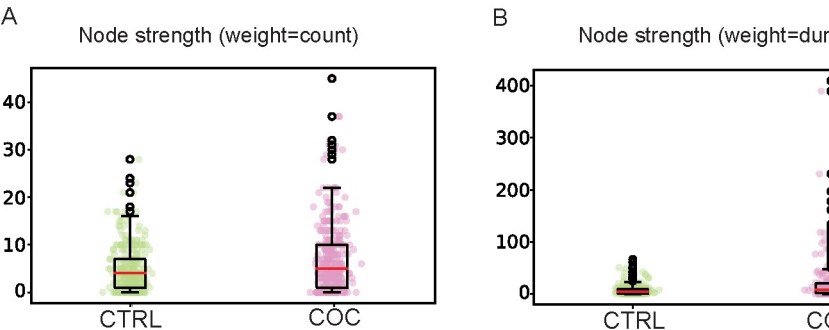

**Fig 5. Information centrality indicate smaller number of hubs in CTRL networks. A)** Box plot of median values for nine CTRL networks (n = 270 flies) grouped, which were given regular food, and eleven COC networks (n = 330 flies) grouped, which were orally administrated 0.50 mg/mL of cocaine for 24 hours before tracking. Data are extracted from 10 minute videos using FlyTracker and analyzed using NetworkX. Statistical analysis was performed using independent-samples *t*-tests with Welch-correction since group sizes are different. *p*-value less than 0.05 is taken as significant presented in S1 Table. **B)** Box plot of nine CTRL networks (n = 270 flies) grouped on the regular food and eleven COC networks (n = 330 flies) grouped and orally administrated to 0.5 mg/mL of cocaine for 24 hours before tracking. Data are extracted from 10 minute videos using FlyTracker and analyzed using NetworkX. **C)** Histogram for average information centrality from nine CTRL networks (n = 270 flies) grouped on the regular food. **D)** Histogram for average information centrality from eleven COC networks (n = 330 flies) grouped and orally administrated to 0.5 mg/mL of cocaine for 24 hours before tracking.

node. COC-fed groups that show lower degree centrality also show higher information centrality (Fig 5). CTRL groups had lower average information centrality (Fig 5A and 5C).

Additionally, by analyzing the weighted strength distribution, either for the number of interactions (Fig 6A), or based on the duration of those interactions (Fig 6B), we show that COC-fed flies have higher values than the untreated groups. The difference between CTRL and COC population is larger for the duration of the video recording than for the number of interactions. indicating that there are more interactions in COC groups and that they last longer.

Local level measures of eigenvector centrality (S2 Fig) did not show differences between CTRL and COC fed groups. Similarly, the betweeness centrality showed no significant difference (S8 Fig). Even when we applied weights for duration and count (S9 and S10 Figs) we have not observed significant differences. COC feeding also did not influence the node strength even when we applied weights for duration and count (S6 and S7 Figs).

Finally, we have calculated the Pearson's correlation for each pair of local level measures, for the CTRL and COC group (Fig 7). We did not observed any differences in correlation matrix.

## CTRL networks have better clustering into communities on the middle-level

Our observation of a higher number of isolated individuals among COC fed flies at the local-level is partially confirmed at the middle level of analysis (Table 2). There are isolated flies without any interaction in both CTRL and COC populations, but among COC fed flies there is a trend for a higher, albeit a non-significant number of isolated flies (Table 2).

Each fly represents a node, and nodes can over time tie into communities within the group, that are characterized by their size and permanence. Feeding with COC led to the formation of fewer, but more populated communities, while we observe the opposite in the untreated groups (Table 2). Middle-level analysis based on the weighted measures for duration and count is reported in Table 2 and in the Supporting Information (S1 and S2 Tables). Count

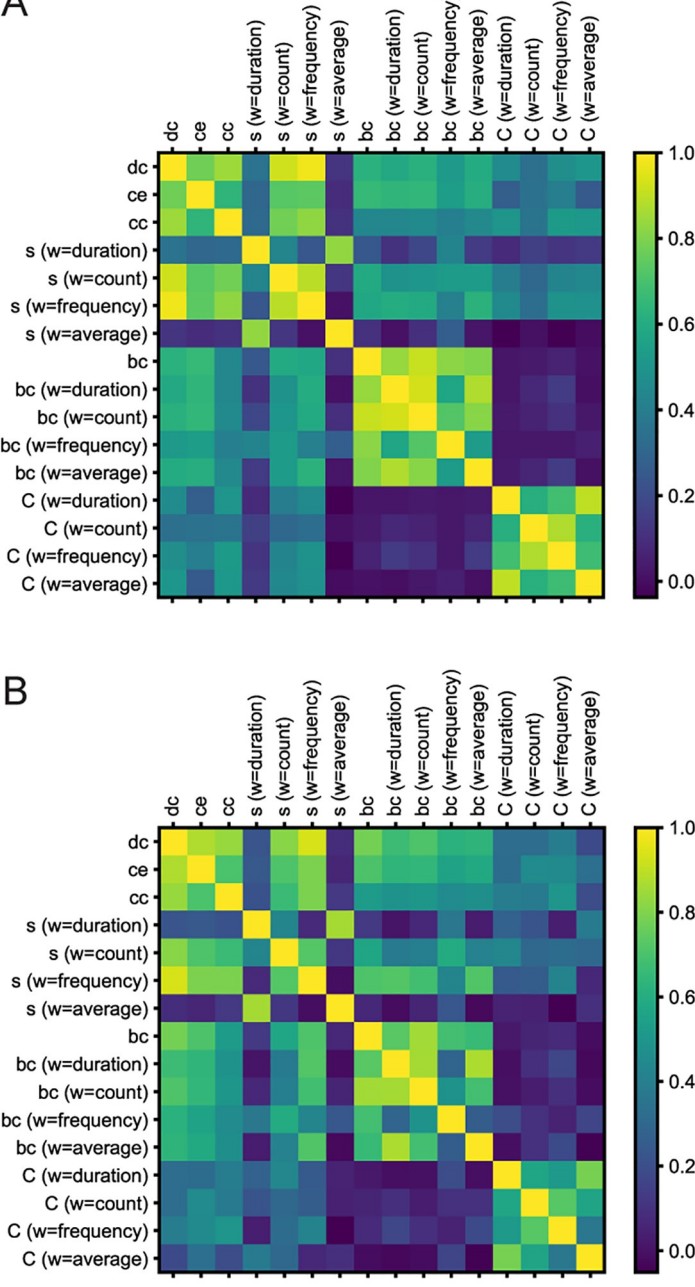

**Fig 6. Node strength distribution using weights uncover more of longer interactions in the COC SINs.** Strength distribution, using weight **A)** count and **B)** duration. Box plot of average values for nine CTRL SINs (n = 270 flies) grouped, which were given regular food, and eleven COC SINs (n = 330 flies) grouped, which were orally administrated 0.5 mg/mL of cocaine for 24 hours before tracking. Data are extracted from 10 minute videos using FlyTracker and analyzed using NetworkX. Statistical analysis was performed using two-sample *t*-tests for a difference in mean for paired samples with a p-value less than 0.05 taken as significant, presented in an S1 Table.

weights non-significantly reduced the number of elements in the biggest communities in COC groups (S1 Table), but such an effect was not observed in the untreated groups (S5 Table). The weighted values for counts did not vary either as s factor of community size or group (S1 Table).

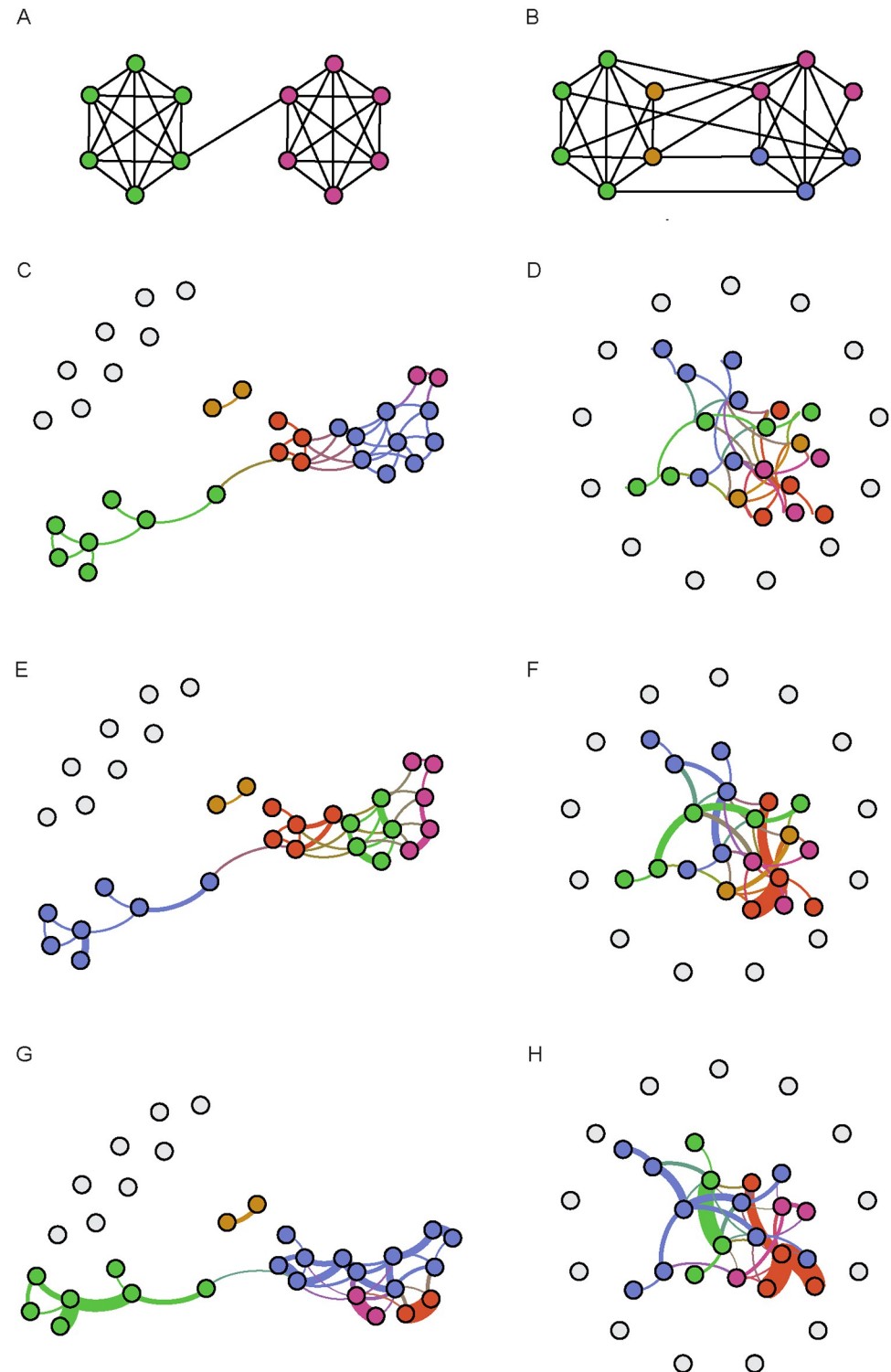

**Fig 7. Correlation matrix.** Image represents correlation of local level measures in created networks. **A)** is for CTRL networks and **B)** is for COC networks. Full names of measures that are abbreviated in the correlation matrix figure are: dc—Degree centrality, ce—Eigenvector centrality, cc—Closeness centrality, s—Strength distribution, bc—Betweenness centrality, C—Clustering coefficient and w—weight.

**Table 2. Mid level network-based measures.** Table represent values calculated on middle (community) level of network for control (CTRL) and coaine (COC) population, where values are calculated using duration as node weight.

| Measure | mean COC | mean CTRL |
| --- | --- | --- |
| Number of nodes | 28±0.74 | 31.11±0.72 |
| Number of single element communities | 4.64±1.16 | 5.67±1.35 |
| Percentage of single element communities | 20.23% | 14.90% |
| Number of communities | 8.63±1.09 | 10.67±1.52 |
| Number of communities without single | 4±0.27 | 5±0.4 |
| Biggest community size | 12.09±1.05 | 8.44±0.81 |
| Second biggest community | 5.18±0.74 | 6.67±0.47 |
| Average community size without single elements | 5.97±0.34 | 5.49±0.59 |
| Number of components | 7.00±1.32 | 5.64±1.28 |
| Biggest component size | 24.78±1.42 | 23.36±1.41 |
| Modularity (Q) | 0.22±0.02 | 0.37±0.04 |

An important distinction between the CTRL and COC populations at the middle-level was in their modularity values (Table 2, S5 Table). CTRL groups have higher modularity values indicating that flies form communities that contain denser internal connections, as opposed to connections outside the community. In the COC groups, there are more flies (nodes) that interact with other communities.

We analyzed several types of weights and links between two nodes at the middle-level: links without weights that describe the number of interactions that one node has with other unique nodes; weighted links that quantify all interactions between any two specific nodes; weighted links that quantify the duration of all interactions between two specific nodes. Examples for three representative communities for CTRL and COC fed flies show that there is consistency among communities that form and that are independent of the type of analysis: unweighted, weighted for the number of interaction, or their duration (Fig 8). Consistent with previous measures at the local and middle level, COC treatment leads to the formation of larger, more densely connected but isolated communities.

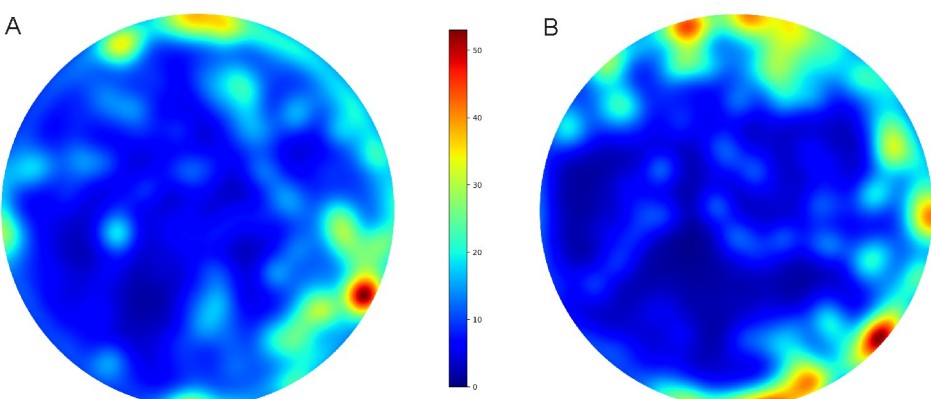

**Fig 8. Higher number and duration of interaction in denser COC communities. A)** Hypothetical network representation with high modularity **B)** Hypothetical network representation with low modularity **C)** Unweighted communities in CTRL **D)** and COC population **E)** weighted networks for number of interactions in CTRL **F)** and COC population, **G)** weighted networks for duration of interactions for selected CTRL **H)** and COC group. Each community is represented in different colour and links of different thickness represent the link weight. Ilustrations are created using the open-source software Gephi.

## Discussion

In this paper we present a network-based methodology for quantitative analysis of social interaction networks (SINs) of *Drosophila* using network-based measures. We analyzed a group of 30 freely moving adult male flies in a circular arena, and analyzed them using network-based measures at the local, middle and global level. To test the neuromodulatory effect of COC we used flies of the same genotype fed with COC, and observed subsequent changes in their network-based measures and SINs. Measures were complementary within each level of analysis and consistent among different levels considering control and COC-fed flies. Within the proposed methodology, we select and calculate a set of network-based measures that provide insight into the network structure and possibly behavioral properties of files represented as nodes within the observed SINs. Specifically, we showed that selected network measures can differentiate between COC and CTRL *Drosophila melanogaster* populations. We implemented the proposed methodology using NetworkX and utilized this approach to detect cocaine-induced changes in social interactions in *Drosophila melanogaster*.

Individual patterns of social interaction relate to individual and group characteristics and can be analyzed using standardized mathematical methods for calculating metrics of socialization in interaction networks of animals [48]. Application of network-based measures to describe SINs can be used to find patterns of behavior within groups of individuals in different species [49–51], observe changes in group relationships under external or internal influence [28] and understand behavior on a wider scale. Advances in bioinformatics have enabled studies of the social behavior in large groups of *Drosophila*, which now makes it possible to combine genetic approaches and environmental manipulation to define neurobiological mechanisms that govern social interaction.

The methodology for acquisition, analysis, and graphical representation of social interaction in flies, as well as the terminology used to describe social interactions, is not always consistent and semantically clearly defined. This is a consequence of different approaches being applied to define this complex behavior and analyze its biological underpinning. However, multiple studies agree in their conclusions that flies show an innate ability to interact with one or more individuals and that over time this can result in the formation of interacting dyads, groups, clusters, or social interaction networks [19, 52, 53].

Our study uses network-based analysis that has previously proven to be successful for describing social interaction [21, 28]. To the best of our knowledge, this is the first attempt to apply network-based analysis at the local, middle, and global levels to detect cocaine-induced changes in *Drosophila* social interaction networks.

At the local level, we analyzed centrality measures describing the behavior of individual flies, and their possible influence on other flies and the network. These measures can be weighted for the duration, frequency, number, or other variables, in order to provide a precise characterization of social interactions and insight into networks. In the context of network-based analysis, local level centrality measures can be used in the identification of the most important flies in the network. Depending on the context, the most important fly in the network can be defined as the one with a high number of interactions with other flies, then it is important to choose the centrality measure which involves this criterion.

Centrality measures can be used to make inferences about information flow. In our study, we did not directly measure information flow, although it was shown that flies transfer information during physical interaction, such that "uninformed" flies make the decision to follow or not to follow the "informed" flies [27]. Based on measured centrality values, we propose potential scenarios regarding centrality measures and information flow.

At our local level, the network-based analysis showed that untreated flies had a higher degree centrality and clustering coefficients. These two measures agree since the first indicates interaction originating from a single fly (node), and the second shows that subgroups or cliques form as a result of interactions. A higher degree may imply popularity. Thus, in the context of fly SINs, we speculate that a higher number of interactions among untreated flies and the formation of subgroups ultimately led to a higher number of popular flies (nodes) and a lower number of isolated flies, compared to COC-fed flies (Fig 4H). COC fed flies showed increased closeness and information centrality. This indicates that interacting flies in a group are tightly spaced, and consequently potential information flow is higher because information flow in this context is the inverse of path length. Furthermore, COC-fed flies had higher node strength weighted in both number and duration indicating a higher number and longer duration of interactions. In terms of information flow, this measure leads to contradicting explanations. The higher number of interactions suggests a faster spreading of information, however, the longer duration of interactions slows it down.

Oral administration of methamphetamine, a psychostimulant with similar molecular and behavioral effects as cocaine, leads to increased sexual arousal, longer duration of courtship sequences, and decreased number of copulatory endpoints, suggestive of the males' inability to adapt their behavior to signals they receive from females [54]. It is possible that the longer duration of interaction between COC fed males represents a similar form of repetitive behavior where flies persist in interactions and are unable to modify their behavior according to the signals they receive from the other fly. Due to the resolution of our recording set-up, and the definition of interaction between two flies, we were not able to define the type of behaviors that males performed, although if COC led to similar increased sexual arousal as methamphetamine this may result in male-to-male courtship. Male to male courtship is well described in *Drosophila* and is a finely orchestrated sequence of behaviors that are governed by the genetic landscape, internal state and mostly chemosensory cues [55–57].

In both flies and mice, a small group of neurons controls mating and aggression that can be behaviorally observed as interspersed mating and aggressive sequences between males [58, 59]. In *D. melanogaster* males, a male-specific group of P1 fruitless neurons in the central brain promote either mating or aggression, depending on the level of optogenetic stimulation and activation [59]. In our experiment, flies were housed for 24 hours on a food substrate that contained 0.50 mg/mL COC, where it is not possible to control the amount of food intake. Depending on the amount of ingested food and individual sensitivity to the arousing effects of COC, it is likely that this resulted in a range of behaviors, such as short sequences of mating and/or aggression, since these two innate behaviors are triggered in an inverse manner that is threshold-dependent. This explanation can be tested in the future using experimental approaches that allow for control of the amount of food ingested by individual flies [13] and the use of higher resolution cameras for video recording.

The same group of neurons modulates persistent internal states that can be referred to as motivation, arousal, or drive that is required for mating and aggression [58], but are also involved in the formation of social interaction networks, such as the ones measured in our study. Importantly, P1 fruitless neurons receive input from neuronal circuits that process pheromonal cues [59]. A number of studies have shown the influence of chemosensory cues on the formation of social interaction networks [21, 26, 52, 53, 60, 61].

Olfactory cues are of great importance since flies with mutations in genes that are required for processing of olfactory cues form disrupted social networks, relative to wild-type controls [52]. A number of genetic mutants for sensory and gustatory processing also showed severe disruption in social clustering, a form of time-dependent social behavior [53]. Thus, the effect that COC feeding has on types of network interactions in our experiment involves

neuromodulation of internal states that can potentially result in the increased arousal or motivational behavior on one side and changed processing of chemosensory cues that result in repetitive behavior resulting in increased duration of interactions between flies.

A recent rodent study shows that social interactions change the activity of specific dopaminergic circuits that control drug craving, suggesting that social interaction has a reinforcing influence on addictive behaviors [62, 63]. The brain of *Drosophila* is comprised of some 150.000 neurons, of which 127 per hemisphere are dopaminergic (DA) and 80 serotonergic (5-HT) [64, 65]. Their activity regulates behavioral functions equivalent to those in mammals, such as motor activity, reward and aversion, memory formation and feeding [66]. It has been shown that dopaminergic neuronal activity influences social behavior and that individual social background affects local, middle, and global parameters in group behavior. COC inhibits the reuptake of DA from synaptic clefts by binding to the dopamine transporter on the presynaptic neuron membrane and increases the concentration of free extracellular DA. COC can also enter the neurons since it is a lipophilic weak base with a positive charge at physiological pH. COC causes intracellular mitochondrial impairment, and increased levels of DA since it binds to the vesicular monoamine transporter (VMAT) [67, 68]. It is known that mutants in *Drosophila* DAT named *fumin* (*fmn*) and VMAT have opposing effects on locomotion. *fmn* mutant flies are hyperactive [69] while VMAT mutant flies have lover activity [70]. On the other hand, overexpression of VMAT leads to increased locomotor activity in flies and altered response to cocaine [71]. Since we have observed an increased number of isolated individuals in the COC population, it could be a result of different mechanisms regulating DA reals inside and outside of dopaminergic neurons, and a possible consequence of different frequency and duration of COC feeding in different flies resulting in different COC concentrations.

The results of network analysis at the middle and global level of the analysis confirm results obtained at the local level. At the middle network level of analysis, untreated flies show lower modularity. The communities that form have a moderate level of connections within the group, but they have nodes that interact with neighboring communities (Fig 8). In COC-fed flies, we observe a similar effect as at the local level with denser communities with a higher number and duration of interaction within the community and without connection to other communities. Thus, although the COC networks seem to be better connected in terms of information centrality, the quality of connections is better in the CTRL networks. At the global level, we have the least significant differences between CTRL and COC fed flies. However, we saw the recapitulation of previously reported measures, such as the size and density of the social groups that encompassed a smaller diameter where flies had longer interactions. Another important difference that is evident at all levels was the larger number of isolated flies among those that consumed COC. A potential explanation for this effect is unevenness in COC dosing. Depending on the dose, COC feeding can lead to increased arousal and motor activation, but above a certain threshold, it can lead to decreased activity or paralysis [10]. An alternative explanation is that flies were recuperating from excessive stimulation during 24-hour exposure on COC supplemented food substrate that then led to decreased arousal and potentially sleeping. The large, dense groups that were seen among COC-fed flies are reminiscent of the recently described time-dependent social cluster formation [53]. Over 120 minutes, flies spontaneously progress from loosely connected communities to a single social cluster positioned by the edge of a circular arena. At 10 minutes, which is the length of our recording, Lifen et al. report that wild-type flies form groups similar to the ones present in our untreated group. However, COC fed flies formed tight groups akin to the clusters that form after 120 minutes in untreated flies [53]. Clustering evolves from dyadic interaction between flies through touch events, and the critical component is the activity of the ppk gene coding for ion channels involved in mechanosensation [53]. Inactivation of ppk prevents the formation of

social clusters, while activation decreases the time for clustering. This gene is expressed in the nervous and sensory system, thus the function of the ion channel for which it codes is likely to be affected in COC fed flies, such that heightened arousal leads to either an increased activity of this channel or to the higher frequency of touch events resulting in the formation of more populated groups that we observe after 10, instead of 120 minutes.

## Conclusion

In this paper we propose a novel network-based methodology for quantitative analysis of social interaction networks of *D. melanogatser*. We implement the methodology using the NetworkX library enabling the efficient analysis of the network structure at the local, middle and global levels. Implementation of the proposed approach in NetworkX provides a novel level of analysis and insight into social interactions of *D. melanogatser*. We have successfully applied methodology in the task of analysis of social interaction networks. To validate the proposed methodology, we compared control (CTRL) and cocaine (COC) 0.50 mg/mL fed flies global-level network-based measures in locomotion. We propose a novel set of network-based measures at each level of analysis that demonstrates the influence of COC on groups of *Drosophila* males. A substantial number of the parameters measured differed significantly between these two populations. Differences were most pronounced at the local level of analysis, but also showed consistency in the other levels of analysis. The relevance of our findings is twofold: we have shown that cocaine-induced effects on brain physiology led to changes in social interactions, and secondly, we were able to explain and support cocaine-induced changes in social behaviors in the context of changes in sensory processing and brain functioning that cocaine is known to induce. These explanations suggest new hypotheses that can in the future be tested using a more refined methodological approach.

## Supporting information

**S1 Protocols. Detailed step-by-step description of the experiment, covering everything from the breeding of Drosophila melanogaster to the construction and analysis of networks.** It can be found at dx.doi.org/10.17504/protocols.io.e6nvwk7q2vmk/v1.
(TXT)

**S1 Fig. Degree centrality.** Box plot graph represents measures of Degree centrality measure distribution across networks in CTRL and COC populations.
(TIF)

**S2 Fig. Eigenvector centrality.** Box plot graph represents measures of Eigenvector centrality measure distribution across networks in CTRL and COC populations.
(TIF)

**S3 Fig. Closeness centrality.** Box plot graph represents measures of Closenes scentrality measure distribution across networks in CTRL and COC populations.
(TIF)

**S4 Fig. Information centrality.** Box plot graph represents measures of Information centrality measure distribution across networks in CTRL and COC populations.
(TIF)

**S5 Fig. Strength (weight = duration).** Box plot graph represents measures of Strength measure distribution across networks in CTRL and COC populations, where weight is duration.
(TIF)

**S6 Fig. Strength (weight = count).** Box plot graph represents measures of Strength measure distribution across networks in CTRL and COC populations, where weight is count.
(TIF)

**S7 Fig. Betweenness centrality.** Box plot graph represents measures of Betweenness centrality measure distribution across networks in CTRL and COC populations.
(TIF)

**S8 Fig. Betweenness centrality (weight = duration).** Box plot graph represents measures of Betweenness centrality measure distribution across networks in CTRL and COC populations, where weight is duration.
(TIF)

**S9 Fig. Betweenness centrality (weight = count).** Box plot graph represents measures of Betweenness centrality measure distribution across networks in CTRL and COC populations, where weight is count.
(TIF)

**S10 Fig. Clustering coefficient (weight = duration).** Box plot graph represents measures of Clustering coefficient measure distribution across networks in CTRL and COC populations, where weight is duration.
(TIF)

**S11 Fig. Clustering coefficient (weight = count).** Box plot graph represents measures of Clustering coefficient measure distribution across networks in CTRL and COC populations, where weight is count.
(TIF)

**S1 Table. Mid level results (weight = duration).** Statistical values from networks over mid-level measures calculated using duration as weight.
(XLSX)

**S2 Table. Mid level results (weight = count).** Statistical values from networks over mid-level measures calculated using count as weight.
(XLSX)

**S3 Table. Global level results.** Statistical values from networks over global level measures.
(XLSX)

**S4 Table. T-test for local measures.**
(XLSX)

**S5 Table. T-test for middle level measures.**
(XLSX)

**S6 Table. T-test for global measures.**
(XLSX)

## Author Contributions

**Conceptualization:** Milan Petrović, Ana Meštrović, Ana Filošević Vujnović.

**Data curation:** Milan Petrović, Ana Filošević Vujnović.

**Formal analysis:** Milan Petrović, Ana Filošević Vujnović.

**Funding acquisition:** Rozi Andretić Waldowski, Ana Filošević Vujnović.

**Investigation:** Milan Petrović.

**Methodology:** Milan Petrović, Ana Meštrović, Rozi Andretić Waldowski, Ana Filošević Vujnović.

**Software:** Milan Petrović, Ana Meštrović.

**Supervision:** Ana Meštrović, Ana Filošević Vujnović.

**Validation:** Rozi Andretić Waldowski.

**Visualization:** Milan Petrović.

**Writing – original draft:** Milan Petrović, Ana Meštrović, Ana Filošević Vujnović.

**Writing – review & editing:** Rozi Andretić Waldowski.

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
