## [Decision Letter · Decision Letter 0]

30 May 2022

PONE-D-22-09891A network-based analysis detects cocaine-induced changes in social interactions in Drosophila melanogasterPLOS ONE

Dear Dr. Petrović,

Thank you for submitting your manuscript to PLOS ONE. After careful consideration, we feel that it has merit but does not fully meet PLOS ONE’s publication criteria as it currently stands. Therefore, we invite you to submit a revised version of the manuscript that addresses the points raised during the review process.

Per the attached detailed reviews, both authors find the manuscript meritorious, but raise concerns (see reviewer 1) that do merit toning down a bit the interpretation of the results. Some methodological questions were raised as well that need to clarified. I agree with reviewer 1, bringing up the issue of terminology and jargon used. Since the terminology used to describe terms is generally not accessible to the broad scientific audience of the Journal, it needs to be explained in more "user friendly" terms, or at least minimize the field-specific jargon.  Please thoroughly address these concerns and comments. 

We look forward to receiving your revised manuscript.

Kind regards,

Efthimios M. C. Skoulakis, PhD

Academic Editor

PLOS ONE

Journal Requirements:

2. Please ensure you state in the Materials section of your manuscript text the origin (supplier or manufacturer) of the cocaine used in this study.

Reviewers' comments:

Reviewer's Responses to Questions

**Comments to the Author**

1. Is the manuscript technically sound, and do the data support the conclusions?

Reviewer #1: Yes

Reviewer #2: Yes

2. Has the statistical analysis been performed appropriately and rigorously? 

Reviewer #1: No

Reviewer #2: Yes

3. Have the authors made all data underlying the findings in their manuscript fully available?

Reviewer #1: Yes

Reviewer #2: No

4. Is the manuscript presented in an intelligible fashion and written in standard English?

Reviewer #1: No

Reviewer #2: Yes

5. Review Comments to the Author

Reviewer #1: The article by Petrovic et al. describes a novel analysis package, NetworkX, to analyze interactions amongst 30 male flies, as captured by video tracking. Authors extract a (large) number of measured parameters, some of which change in flies that have been fed cocaine for a day. I am unconvinced of the authors interpretations as to the meaning of the data, and the data differences. These (overinterpreted) statements thus should go into the discussion, and not the results and methods. Other than that, the paper is solid, requires some textual edits, and will thusly be in good shape.

Major interpretative issues:

Exposure to cocaine does not “validate” the findings on the myriad of parameters with NetworkX, since it is NOT clear – a priori – what the expected outcomes should be. It was maybe a missed opportunity to actually validate the software with mutants that have been described before in the triangle social interaction assay, e.g. Nlg3. Or say some setups with predictable interactions, all males, vs 15 males with 15 virgins. In the absence of that, the software may very well extract reproducible data (but see Fig5B), but what those data actually mean, is a different (and unanswered) question. Therefore, some of the writing needs to be toned down, specifically:

“[abstract] Larger and denser communities of COC treated flies can be explained by heighten arousal and repetitive behaviors.” Heightened arousal maybe, but I can’t recall repetitive social stereotypies being described in the Drosophila cocaine exposure literature. Sounds like conjecture that can go into the discussion, but not into the abstract.

“[abstract] Our work showed that social network analysis using NetworkX identifies biologically relevant parameters that can be used…” Relevant for what? Is the definition of relevant here: to be measurable and affected by cocaine?

“l.277 because information in this context is an inverse to the path length” to a social behavioral biologist information has little to do with path length.

“l.291 indicating that nodes had no influence on the flow of information in the network” what is the definition of “flow of information” here?

“l.345 Ultimately, this leads to higher global efficiency in the COC relative to CTRL networks.” What does efficiency mean here?

Second: the paper needs to be a bit more accessible to general readers. The problem is that the manuscript marries two distant fields: engineering/data science with behavioral biology. Each of these fields has its own jargon, or even definition of certain words: “information flow” in a system does not have the same implications as “information flow” between sentient beings. Thus authors need to keep naïve readers from EITHER discipline in mind, and they also need to precisely define what they mean with certain key terms.

Third: It’s a bit odd to see all these t-tests (in the supplement), when the data is displayed with medians and quartiles. Seems to require non-parametric testing and some adjusting of the p-value (with so many comparisons). Also, why paired tests (given there were 9 ctl, but 11 coc experiments)?

Minor:

Abstract: “Importantly the differences were complementary within each level of analysis and consistent among different levels.” not sure this is well explained in the text.

Fig1D legend: is standing away 4mm from another fly really an interaction? Why 4mm, why not 3 or 5, or 0, really? Was this parameter explored and determined heuristically, or is this based on some assumption, or some relevant published data? And where do these 4mm start/end? At the centroid? At the periphery facing the interactor (as suggested by the figure). Then again, in the methods it says “focused on touch”. Confusing!

Is a node a fly? The same fly followed over 10 min? a snapshot in time? (actually, nodes represent the flies, not the other way around l.115)

l.119: “For each network we introduced two variants of weight factors:..” is this supposed to mean: for each interaction we measured two variables…represented in the network as…

the rest is also confusing. What if two flies make contact in minute 2, and again in minute 8. How is that represented? Same link? Thus l.121 should read “number of times the same flies made contact during the 10-min observation period…”?

L126: “In following subsections we provide definitions and interpretations of the..” Definitions, yes; interpretations: no. Interpretations go into the discussion, since I see no evidence provided that centrality equates to “the most influential individuals”. That’s too much anthropomorphizing! Same with “which nodes control the network (in terms of information flow)”. There is no evidence a specific fly is socially ‘dominant’ in controlling information flow. I posit that if you take 30 marbles and gently shake them, the analysis will find some marbles that are more dominant, just by chance. Considerably more work is required to infer any meaning into the data as authors are doing, the easiest of which would have been to ask how reproducible are these networks are in a 30 min movie of the same flies analyzed as 3x10 min. IS there really a reproducibly popular, or ‘dominant’ node? ARE there reproducible assortative groups?

Fig. 3A: the median is lower in coc, but the quartiles are higher. How can that really be interpreted meaningfully? Also, it says “average” in the legend, which normally means “mean”, but such details are not explained cogently. And how do you pair these samples!?! Each dot is a node=fly, and each ctl fly has a sibling fly in the coc group??? That makes no sense…

Fig4B gives one pause regarding reproducibility, especially the ctl side.

l.289: “signifying that nodes have no influence on the network” huh? The network consists of linked nodes, how can they NOT have an effect? Unclear

l.290: “SINs does not influence flies thought the incoming links” that’s not English, but also, why are we talking about flies now, not nodes?

Reviewer #2: This article is the validation of a new technique to quantify Social Interaction Networks (SIN) in flies, assessing the effect of feeding cocaine to the flies on emerging properties of SINs, in a group of 30 males, which had never been done before.

It is well written and easy to follow. The methodology is mostly clearly described (see comment below). The figures are well done, especially Fig1. The statistical analysis appears fair to me, the software used to analyze the video is made freely available. I particularly appreciated the fact that the authors describe very clearly the emerging properties, at 3 different levels (local, middle and global), and their interpretations.

Major concerns:

More context is needed. The new methodology presented here should be compared and contrasted to other reported techniques used to quantify SIN in flies – see recent review by Jezovit JA, Alwash N and Levine JD (2021) Using Flies to Understand Social Networks. Front. Neural Circuits 15:755093.doi: 10.3389/fncir.2021.755093

Methodology: well described, apart for the rationale behind the cocaine feeding section.

Why feeding only 24 hours? Why 0.5 mg/ml? Ref?

Discussion:

Overall, discussed different aspects and interpretation of the results presented. I have just one concern here. The authors discuss how sensory modalities, known to be important for SINs properties are received by FruP1 neurons, which in turn promote aggression or courtship. They also explain that the differences in the emerging properties of the flies fed cocaine versus control could be the result of increased arousal, similar to feeding methamphetamine. Also ppk is discussed. However, a whole section on the molecular targets of cocaine (DAT, VMAT), and how behaviours are known to be affect in flies after administering cocaine, or when the encoding genes are mutants is missing. Indeed, not only social behaviours are affected, but also locomotion, grooming etc… Those could also interfere with SINs, not only repetitive behavioural loops.

Minor issues:

The raw data have not been available – only the software

Editorial suggestions

• The overall text is great, nothing particular jumped at me. But the abstract could be more specific:

*Fruit fly, D. melanogaster is a versatile genetic organism with high homology with vertebrates, including humans.*

o PLOSOne is for a non-specialist audience: Homology at what level?

*complex behaviors that can be studied in Drosophila it has become an excellent model *

o I would add a comma before “it”

*Our first aim was to apply NetworkX software for quantitative analysis of social interaction networks on the local, middle and global level. Secondly, to validate our findings we compared those to social interaction in cocaine (COC) treated males *

o Before speaking of aims – I would tell the reader what the research question was. And what findings are the authors trying to validate?

• The first time in the paper than Drosophila melanogaster, it should be written in full. After that, either write consistently the full name, or shorten it to D. melanogaster, but be consistent. Also, it should always be in italics.

• Reference list: the format is not homogeneous (especially for the titles: sometimes each word start with Caps, sometimes not, but also websites listed for some but not all of the articles).

6. PLOS authors have the option to publish the peer review history of their article (what does this mean?). If published, this will include your full peer review and any attached files.

Reviewer #1: No

Reviewer #2: **Yes: **Anne F. Simon

---

## [Author Response · Author response to Decision Letter 0]

6 Aug 2022

Rebuttal Letter

We thank the editor and the two reviewers for their reviews and list of constructive

suggestions. We are very grateful for the opportunity to submit a major revision of our paper

for your renewed consideration. Below is our response to each point raised by the academic

editor and reviewers. We hope that we satisfyingly addressed them satisfactorily and that the

manuscript is now suitable for publication.

Editor

1. Per the attached detailed reviews, both authors find the manuscript meritorious, but raise

concerns (see reviewer 1) that do merit toning down a bit the interpretation of the results.

Some methodological questions were raised as well that need to clarified. I agree with

reviewer 1, bringing up the issue of terminology and jargon used. Since the terminology used

to describe terms is generally not accessible to the broad scientific audience of the Journal,

it needs to be explained in more "user friendly" terms, or at least minimize the field-specific

jargon.

Please thoroughly address these concerns and comments.

Our response. Dear editor, we have adopted booth reviewers' suggestions and remarks

and rewrote the manuscript minimising field-specific jargon.

2. If applicable, we recommend that you deposit your laboratory protocols in protocols.io to

enhance the reproducibility of your results. Protocols.io assigns your protocol its own

identifier (DOI) so that it can be cited independently in the future. For instructions

see: https://journals.plos.org/plosone/s/submission-guidelines#loc-laboratory-protocols.

Additionally, PLOS ONE offers an option for publishing peer-reviewed Lab Protocol articles,

which describe protocols hosted on protocols.io. Read more information on sharing protocols

at https://plos.org/protocols?utm_medium=editorialemail&utm_source=authorletters&utm_ca

mpaign=protocols.

Our response. Thank you very much for this suggestion. We have decided to deposit our

protocols in protocols.io.

Journal Requirements

1. Please ensure that your manuscript meets PLOS ONE's style requirements, including

those for file naming. The PLOS ONE style templates can be found at

and

https://journals.plos.org/plosone/s/file?id=ba62/PLOSOne_formatting_sample_title_authors_

affiliations.pdf

Our response. Thank you very much for this remark related to the structure of the text of the

paper and the author's affiliation. We accidentally missed this.

2. Please ensure you state in the Materials section of your manuscript text the origin (supplier

or manufacturer) of the cocaine used in this study.

Our response. Thank you very much for this comment. We inadvertently neglected to

include all the information in the first version of the manuscript. Now we have provide all the

necessary information about the origin of the cocaine used in our study.

3. Please note that PLOS ONE has specific guidelines on code sharing for submissions in

which author-generated code underpins the findings in the manuscript. In these cases, all

author-generated code must be made available without restrictions upon publication of the

work. Please review our guidelines

at https://journals.plos.org/plosone/s/materials-and-software-sharing#loc-sharing-code and

ensure that your code is shared in a way that follows best practice and facilitates

reproducibility and reuse.

Our response. We shared all our tracking data and code package freely online under BSD-3

License, so it can be further used.

Reviewer #1:

1. The article by Petrovic et al. describes a novel analysis package, NetworkX, to analyze

interactions amongst 30 male flies, as captured by video tracking. Authors extract a (large)

number of measured parameters, some of which change in flies that have been fed cocaine

for a day. I am unconvinced of the authors interpretations as to the meaning of the data, and

the data differences. These (overinterpreted) statements thus should go into the discussion,

and not the results and methods. Other than that, the paper is solid, requires some textual

edits, and will thusly be in good shape.

Our response. We thank the reviewer for useful comments and suggestions how to improve

our manuscript. We rewrote all unclear parts of the manuscript and moved statements

related to interpretation to the Discussion section. We agree that certain parts of the

manuscript were over interpreted, we have completely removed these statements and „toned

down“ some of our conclusions.

Major interpretative issues:

2. Exposure to cocaine does not “validate” the findings on the myriad of parameters with

NetworkX, since it is NOT clear – a priori – what the expected outcomes should be. It was

maybe a missed opportunity to actually validate the software with mutants that have been

described before in the triangle social interaction assay, e.g. Nlg3. Or say some setups with

predictable interactions, all males, vs 15 males with 15 virgins. In the absence of that, the

software may very well extract reproducible data (but see Fig5B), but what those data

actually mean, is a different (and unanswered) question.

Our response. This is very important remark, thank you. We now realise that in the first

version of our manuscript we did not clearly explain the validation of the proposed approach

(and software). We rewrote certain parts of the manuscript in order to explain the validation

procedure.

We validate the proposed approach (software/methodology) by comparing network-based

measures of COC and CTRL populations on the global network level.

Since it has already been confirmed by the previous experiments that there are differences in

locomotor activities and behaviour of cocaine-induced D. Melanogaster (McClung C, Hirsh J.

Stereotypic behavioral responses to free-base cocaine and the development of behavioral

sensitization in Drosophila. Current Biology. 1998;8(2):109–112, Filošević A, Al-samarai S,

Andretic R. High Throughput Measurement of Locomotor Sensitization to Volatilized Cocaine

in Drosophila melanogaster. Frontiers in Molecular Neuroscience. 2018;11:25), we expected

that networks of COC would exhibit different properties from the CTRL networks, especially

in values of the network measures based on a number of interaction and distance. Therefore,

we compared 13 network measures on the global level and confirmed expected differences

between COC and CTRL populations. Encouraged by these findings, in the next step we

used local and middle network measures to identify changes in SINs of COC populations.

We agree that validation performed by using mutants is an important procedure and we plan

to do that in future work. However, we believe that the influence of cocaine on the olfactory

sensors is insufficiently investigated.

Additionally, we use only male flies for study of addiction so we did not optimized our method

to be influenced by females, but in the future work we will focused on different social

background to define its influence on social behaviour of individuals in the group.

2. Therefore, some of the writing needs to be toned down, specifically:

“[abstract] Larger and denser communities of COC treated flies can be explained by heighten

arousal and repetitive behaviors.” Heightened arousal maybe, but I can’t recall repetitive

social stereotypies being described in the Drosophila cocaine exposure literature. Sounds

like conjecture that can go into the discussion, but not into the abstract.

“[abstract] Our work showed that social network analysis using NetworkX identifies

biologically relevant parameters that can be used…” Relevant for what? Is the definition of

relevant here: to be measurable and affected by cocaine?

“l.277 because information in this context is an inverse to the path length” to a social

behavioral biologist information has little to do with path length.

“l.291 indicating that nodes had no influence on the flow of information in the network” what is

the definition of “flow of information” here?

“l.345 Ultimately, this leads to higher global efficiency in the COC relative to CTRL networks.”

What does efficiency mean here?

Our response. Thank you for these remarks, we agree with all your suggestions. We

rewrote the abstract and tried to clarify the motivation of this research. We moved texts

related to the interpretation to the Discussion section. We toned down our interpretations and

tried to explain concepts from complex network theory, such as path length and global

efficiency. Regarding global efficiency and information flow we agree that these

interpretations and conclusions go too far and therefore we provide a discussion without over

interpretation.

3. Second: the paper needs to be a bit more accessible to general readers. The problem is

that the manuscript marries two distant fields: engineering/data science with behavioral

biology. Each of these fields has its own jargon, or even definition of certain words:

“information flow” in a system does not have the same implications as “information flow”

between sentient beings. Thus authors need to keep naïve readers from EITHER discipline

in mind, and they also need to precisely define what they mean with certain key terms.

Our response. Thank you for this suggestion. Now we rewrote the methodology section (and

all other critical parts of our manuscript) aiming to better explain all technical terms.

Therefore, we provide a more detailed explanation of network-based measures – using

terminology related to the D. melanogaster domain – flies (represented using nodes) and

interactions (represented using links).

4. Third: It’s a bit odd to see all these t-tests (in the supplement), when the data is displayed

with medians and quartiles. Seems to require non-parametric testing and some adjusting of

the p-value (with so many comparisons). Also, why paired tests (given there were 9 ctl, but

11 coc experiments)?

Our response. We corrected that. We now performed statistical analysis using

independent-samples t-tests. These tests are Welch-corrected since group sizes are different

and variance should thus not be assumed equal and significance level of p < 0.05. We

explained that in the paper and correct results in the supplement.

Minor:

5. Abstract: “Importantly the differences were complementary within each level of analysis

and consistent among different levels.” not sure this is well explained in the text.

Our response. Thank you for this remark. We reformulated this sentence and explained

better what we wanted to say.

6. Fig1D legend: is standing away 4mm from another fly really an interaction? Why 4mm,

why not 3 or 5, or 0, really? Was this parameter explored and determined heuristically, or is

this based on some assumption, or some relevant published data? And where do these 4mm

start/end? At the centroid? At the periphery facing the interactor (as suggested by the figure).

Then again, in the methods it says “focused on touch”. Confusing!

Our response. That is a good question. We explained that in more detail in the new version

of our manuscript. The criteria of how many body lengths and what time should be

considered as social interaction are taken from the similar experiment setup published in the

related study that quantified these values (Schneider et al, 2014).

7. Is a node a fly? The same fly followed over 10 min? a snapshot in time? (actually, nodes

represent the flies, not the other way around l.115).

Our response. Yes, flies are represented using nodes. One node represents the same fly

followed in the snapshot time. We tried to clarify that in the new version of our manuscript.

8. l.119: “For each network we introduced two variants of weight factors:..” is this supposed to

mean: for each interaction we measured two variables…represented in the network as…

the rest is also confusing. What if two flies make contact in minute 2, and again in minute 8.

How is that represented? Same link? Thus l.121 should read “number of times the same flies

made contact during the 10-min observation period…”?

Our response. Thank you for that comment. We are aware that in our first submission we

did not clearly explain weights. We thank the reviewer for pointing this out. We rewrote this

paragraph and tried to clarify how we construct a weighted network. If two flies, make contact

in minute 2, and again in minute 8 it is represented using the same link. In one case, the

weight would be 10 (which is the total duration of files interaction) and in the second case,

the weight would be 2, because there were two interactions between these two flies.

9. L126: “In following subsections we provide definitions and interpretations of the..”

Definitions, yes; interpretations: no. Interpretations go into the discussion, since I see no

evidence provided that centrality equates to “the most influential individuals”. That’s too much

anthropomorphizing! Same with “which nodes control the network (in terms of information

flow)”. There is no evidence a specific fly is socially ‘dominant’ in controlling information flow.

I posit that if you take 30 marbles and gently shake them, the analysis will find some marbles

that are more dominant, just by chance. Considerably more work is required to infer any

meaning into the data as authors are doing, the easiest of which would have been to ask how

reproducible are these networks are in a 30 min movie of the same flies analyzed as 3x10

min. IS there really a reproducibly popular, or ‘dominant’ node? ARE there reproducible

assortative groups?

Our response. We thank the reviewer for the suggestions regarding the interpretation of

SINs measures. We agree that there is no evidence for our interpretations claimed in these

subsections. We rewrote parts of the section "Characterisation of SINs" and removed all

attempts to interpret the network measures. This way, some of the interpretations are deleted

and some are moved to the Discussion section.

10. Fig. 3A: the median is lower in coc, but the quartiles are higher. How can that really be

interpreted meaningfully? Also, it says “average” in the legend, which normally means

“mean”, but such details are not explained cogently. And how do you pair these samples!?!

Each dot is a node=fly, and each ctl fly has a sibling fly in the coc group??? That makes no

sense…

Our response. Fig. 3 (in the new version it is Fig. 4) illustrates differences between CTRL

and COC SINs in three local network measures: degree centrality, closeness centrality and

clustering coefficient. In this figure each dot is a node=fly, however, there are no „siblings“

because we do not have paired samples of SINs. We compared values of local network

measures of flies from both populations illustrating separately COC population and CTRL

population.

As you said, in Fig. 3A (Now it is Fig, 4A): the median is lower in COC populations, but the

quartiles are higher. This can be explained by the fact that there are 20% of isolated flies

(with no interactions) in COC networks are outliers and do not belong to any quartile. At the

same time, these flies are taken into account when calculating the median..

11. Fig4B gives one pause regarding reproducibility, especially the ctl side.

Our response. This is a good question. Here we really have an outlier in the CTRL

population with higher values of information centrality measure. It may seem that this would

be a problem for reproducibility. Some similar studies treat these situations in the way that

they remove outliers from the population. We decided to keep this outlier in order to show

that in some groups of flies it is possible to have different behaviour. Since the other eight

groups in the CTRL populations have similar values of network measures, we concluded that

in general SINs of the CTRL population have lover values of information centrality.

11. l.289: “signifying that nodes have no influence on the network” huh? The network consists

of linked nodes, how can they NOT have an effect? Unclear

Our response. Thank you for noticing this mistake. We rewrote that paragraph and deleted

this mistake.

12. l.290: “SINs does not influence flies thought the incoming links” that’s not English, but

also, why are we talking about flies now, not nodes?

Our response. We rewrote that paragraph and removed all the unclear parts of the text.

Reviewer #2:

This article is the validation of a new technique to quantify Social Interaction Networks (SIN)

in flies, assessing the effect of feeding cocaine to the flies on emerging properties of SINs, in

a group of 30 males, which had never been done before.

It is well written and easy to follow. The methodology is mostly clearly described (see

comment below). The figures are well done, especially Fig1. The statistical analysis appears

fair to me, the software used to analyze the video is made freely available. I particularly

appreciated the fact that the authors describe very clearly the emerging properties, at 3

different levels (local, middle and global), and their interpretations.

Major concerns:

1. More context is needed. The new methodology presented here should be compared and

contrasted to other reported techniques used to quantify SIN in flies – see recent review by

Jezovit JA, Alwash N and Levine JD (2021) Using Flies to Understand Social Networks.

Front. Neural Circuits 15:755093.doi: 10.3389/fncir.2021.755093

Our response. We appreciate this suggestion. We have included this study in our

manuscript and compared it with our approach.

2. Methodology: well described, apart for the rationale behind the cocaine feeding section.

Why feeding only 24 hours? Why 0.5 mg/ml? Ref?

Our response. Thank you for this remark. We somehow failed to write all these details, but

we agree that this is important. We added all details about cocaine feeding. To ensure the

complete reproducibility of our experiment, we have deposited the entire procedure and

experimental setup at protocols.io. Reason for using 0.50 mg/mL of cocaine-HCl was based

on previous work in our laboratory and other related published results . Since cocaine half-life

is about 1.5 hours and flies were not starved priori of COC feeding, we want to ensure that

concentration used in experiment won’t be aversive to the flies since cocaine is bitter tasting

substance. To be sure, that all flies have consumed COC we left them for 24 hours on food

with COC and that does only increase locomotor activity and not induce other stereotypic

behaviour associated with high doses (Rigo F, Filošević A, Petrović M, Jović K, Andretić

Waldowski R. Locomotor sensitization modulates voluntary self-administration of

methamphetamine in Drosophila melanogaster. Addiction Biology. 2021;26(3):e12963.)

3.Discussion:

Overall, discussed different aspects and interpretation of the results presented. I have just

one concern here. The authors discuss how sensory modalities, known to be important for

SINs properties are received by FruP1 neurons, which in turn promote aggression or

courtship. They also explain that the differences in the emerging properties of the flies fed

cocaine versus control could be the result of increased arousal, similar to feeding

methamphetamine. Also ppk is discussed. However, a whole section on the molecular

targets of cocaine (DAT, VMAT), and how behaviours are known to be affect in flies after

administering cocaine, or when the encoding genes are mutants is missing. Indeed, not only

social behaviours are affected, but also locomotion, grooming etc… Those could also

interfere with SINs, not only repetitive behavioural loops.

Our response. Thank you for this remark. We have add section about influence of cocaine

on the dopamine transporters on presynaptic neurons and inside of neurons on vesicular

monoamine transporter. We discussed influence of those protein targets on locomotor

behaviour and sensitivity to acute dose of cocaine using Drosophila model. We also cited

recent rodent study which shows that social interactions changes the activity of specific

dopaminergic circuits that control drug craving, suggesting that social interaction has

reinforcing influence on addictive behaviours.

Minor issues:

4. The raw data have not been available – only the software

Our response. Thank you for this remark. We made all our data and softer available for

further research.

Editorial suggestions

5. The overall text is great, nothing particular jumped at me. But the abstract could be more

specific:

*Fruit fly, D. melanogaster is a versatile genetic organism with high homology with

vertebrates, including humans.*

o PLOSOne is for a non-specialist audience: Homology at what level?

*complex behaviors that can be studied in Drosophila it has become an excellent model *

o I would add a comma before “it”

*Our first aim was to apply NetworkX software for quantitative analysis of social interaction

networks on the local, middle and global level. Secondly, to validate our findings we

compared those to social interaction in cocaine (COC) treated males *

o Before speaking of aims – I would tell the reader what the research question was. And

what findings are the authors trying to validate?

Our response. Thank you for all these remarks. We completely rewrote the abstract and the

Introduction section. We explained motivation and added a research question in the

Introduction section.

• The first time in the paper than Drosophila melanogaster, it should be written in full. After

that, either write consistently the full name, or shorten it to D. melanogaster, but be

consistent. Also, it should always be in italics.

Our response. We corrected that.

• Reference list: the format is not homogeneous (especially for the titles: sometimes each

word start with Caps, sometimes not, but also websites listed for some but not all of the

articles).

Our response. We checked and corrected the reference list.

---

## [Decision Letter · Decision Letter 1]

7 Sep 2022

PONE-D-22-09891R1A network-based analysis detects cocaine-induced changes in social interactions in Drosophila melanogasterPLOS ONE

Dear Dr. Petrović,

Thank you for submitting your manuscript to PLOS ONE. After careful consideration, we feel that it has merit but does not fully meet PLOS ONE’s publication criteria as it currently stands. Therefore, we invite you to submit a revised version of the manuscript that addresses the points raised during the review process.

As you see the reviews are positive, but because of the uniqueness of this approach it is strongly recommended that the suggestion to explain in more detail the need for this novel approach in the introduction is followed, as well as the other minor editorial suggestions...

Please submit your revised manuscript ASAP. If you will need more time than this to complete your revisions, please reply to this message or contact the journal office at plosone@plos.org. Please include the following items when submitting your revised manuscript:A rebuttal letter that responds to each point raised by the academic editor and reviewer(s). You should upload this letter as a separate file labeled 'Response to Reviewers'.A marked-up copy of your manuscript that highlights changes made to the original version. You should upload this as a separate file labeled 'Revised Manuscript with Track Changes'.An unmarked version of your revised paper without tracked changes. You should upload this as a separate file labeled 'Manuscript'.If applicable, we recommend that you deposit your laboratory protocols in protocols.io to enhance the reproducibility of your results. Protocols.io assigns your protocol its own identifier (DOI) so that it can be cited independently in the future. For instructions see: https://journals.plos.org/plosone/s/submission-guidelines#loc-laboratory-protocols. Additionally, PLOS ONE offers an option for publishing peer-reviewed Lab Protocol articles, which describe protocols hosted on protocols.io. Read more information on sharing protocols at https://plos.org/protocols?utm_medium=editorial-email&utm_source=authorletters&utm_campaign=protocols.

We look forward to receiving your revised manuscript.

Kind regards,

Efthimios M. C. Skoulakis, PhD

Academic Editor

PLOS ONE

Journal Requirements:

Reviewers' comments:

Reviewer's Responses to Questions

**Comments to the Author**

1. If the authors have adequately addressed your comments raised in a previous round of review and you feel that this manuscript is now acceptable for publication, you may indicate that here to bypass the “Comments to the Author” section, enter your conflict of interest statement in the “Confidential to Editor” section, and submit your "Accept" recommendation.

Reviewer #2: (No Response)

2. Is the manuscript technically sound, and do the data support the conclusions?

Reviewer #2: Partly

3. Has the statistical analysis been performed appropriately and rigorously? 

Reviewer #2: Yes

4. Have the authors made all data underlying the findings in their manuscript fully available?

Reviewer #2: Yes

5. Is the manuscript presented in an intelligible fashion and written in standard English?

Reviewer #2: No

6. Review Comments to the Author

Reviewer #2: General comments

Most of the reviewers' comments were appropriately addressed, and the manuscript is mostly ready for publication.

However, I still think that the novelty of the methodology is still not shown. In the introduction the authors still do not explain why a new methodology is necessary. What is the problem the authors are trying to solve? And in the discussion, it is still unclear to me in what is new and advantageous in the method proposed by the authors compared to what was done before. Does NetworkX lead to less errors than Ctrax? Is it easier to use? Does it allow more levels of analysis? A direct clear comparison is necessary.

Edits: I suggest running a spelling and grammar check before final submission, as I picked some typos in the abstract and introduction, but stopped checking there. ,

Third sentence abstract: “we have applied local, middle and global level”: add comma after “middle”

Firth sentence: “and formed larger, densely populated and compact communities” : add comma after “compact”

Last sentence of abstract: Our approach can be expandED on

L2: I propose: “Drug addiction is a complex…”

L6: remove the comma after “abuse”

L10: “This events perturb”  these events perturb

L16: “genES pleiotropy” and “”difficult to assignee”

L20: “metric propensity”  propensities

L25: “This approach maskS”

L32: “transLational

L40: “neuronal citrus analysis” � what do you really want to say?

L75: “by using A library…”

L493: reference missing

7. PLOS authors have the option to publish the peer review history of their article (what does this mean?). If published, this will include your full peer review and any attached files.

Reviewer #2: No

---

## [Editor Report · Decision Letter 2]

26 Sep 2022

A network-based analysis detects cocaine-induced changes in social interactions in Drosophila melanogaster

PONE-D-22-09891R2

Dear Dr. Petrović,

We’re pleased to inform you that your manuscript has been judged scientifically suitable for publication and will be formally accepted for publication once it meets all outstanding technical requirements.

Kind regards,

Efthimios M. C. Skoulakis, PhD

Academic Editor

PLOS ONE
---

## [Editor Report · Acceptance letter]

7 Oct 2022

PONE-D-22-09891R2 

A network-based analysis detects cocaine-induced changes in social interactions in *Drosophila melanogaster*

Dear Dr. Petrović:

I'm pleased to inform you that your manuscript has been deemed suitable for publication in PLOS ONE. Congratulations! Your manuscript is now with our production department. 

Kind regards, 

on behalf of

Dr. Efthimios M. C. Skoulakis 

Academic Editor

PLOS ONE